# Autonomic nervous system modulation by G protein-biased mu-opioid receptor agonists: A translational scoping review protocol

Yi Zhang[1☯], Qiuxiang Chen[2☯], Yuanyuan Zhou[3☯], Qingjun Zeng[1], Haishan Cui[1], Yukai Zhou[4]*

1 Department of Anesthesiology, Wanzhou District Maternal and Child Health Hospital, Chongqing, China, 2 Department of Anesthesiology, People's Hospital of Fengjie, Chongqing, China, 3 Department of Anesthesiology, Guanghan People's Hospital, Sichuan, China, 4 Department of Anesthesiology, Chengdu First Orthopaedic Hospital, Chengdu, China

☯ These authors contributed equally to this work.
* zhouyukai66@outlook.com

## Abstract

### Background

G protein-biased mu-opioid receptor (MOR) agonists such as oliceridine and tegileridine were developed to reduce opioid-related side effects while keeping analgesic efficacy. The US FDA approved oliceridine in 2020 and China NMPA approved tegileridine in January 2024. Clinical development mainly focused on respiratory depression and gastrointestinal problems, but little attention has been paid to the autonomic nervous system (ANS) effects of these drugs. Traditional opioids change heart rate variability (HRV) and sympathovagal balance mainly through central mechanisms. It is not clear whether biased agonists show different autonomic profiles compared with traditional opioids. There is an ongoing scientific debate about whether the benefits of these drugs come from true signaling bias or from low intrinsic efficacy.

### Methods

We will conduct a scoping review following JBI methodology and report according to PRISMA-ScR guidelines. We will search PubMed, Embase, Web of Science and Cochrane Library from database start to March 2026. FDA and NMPA regulatory documents will also be searched. Two reviewers will screen studies and extract data independently. We will include in vitro cellular assays, in vivo animal experiments, and clinical studies examining oliceridine, tegileridine or other biased MOR agonists that report cardiovascular or autonomic outcomes. Results will be organized in an Evidence Matrix by drug, evidence level and measurement timepoint. Animal studies will be stratified by anesthetic state (conscious versus anesthetized).

**Data availability statement:** No datasets were generated or analysed during the current study. All relevant data extracted for this scoping review will be made fully available as Supporting information or via a public repository upon study completion.

**Funding:** This work is supported by the China Red Cross Foundation Medical Empowerment Public Welfare Special Fund 2024 Pain Innovation Action Clinical Research Project. The funder had no role in protocol development and will have no role in conducting, analyzing or publishing the review.

**Competing interests:** The authors have declared that no competing interests exist.

## Discussion

We will map the evidence against three mechanistic paradigms: that biased agonism preserves autonomic homeostasis; that observed benefits come from partial agonism rather than signaling bias; or that no meaningful autonomic differences exist between biased and traditional agonists. This review will map evidence gaps and inform future clinical trial design with autonomic endpoints. We note that tegileridine has much less published data than oliceridine, and this gap analysis will help set research priorities.

**Protocol registration:** Open Science Framework (OSF) [https://doi.org/10.17605/OSF.IO/N2XZC].

## Background

Opioid analgesics remain essential for managing moderate to severe pain, but their use is limited by well-known side effects including respiratory depression, constipation and sedation. For many years, researchers have tried to develop opioids with better safety profiles. One promising approach came from the discovery that G protein-coupled receptors like the mu-opioid receptor (MOR) can activate different intracellular signaling pathways with distinct physiological effects [1]. This phenomenon, called functional selectivity or biased agonism, raised the possibility that drugs could be designed to activate mainly the pathways responsible for pain relief while avoiding those that cause side effects.

The scientific basis for biased MOR agonist development came from studies by Bohn and colleagues in the late 1990s. They found that mice without beta-arrestin-2 had stronger and longer morphine analgesia [2], did not develop tolerance to morphine [3], and showed less constipation and respiratory depression [4]. These findings suggested that G protein signaling produces analgesia while beta-arrestin-2 signaling mainly causes adverse effects.

Based on this hypothesis, several drug discovery programs identified biased MOR agonists. DeWire and colleagues discovered TRV130 (oliceridine), which showed strong analgesia with less gastrointestinal and respiratory problems in animal studies [5]. Manglik and colleagues used computer-based methods to design PZM21, another biased compound [6]. These research efforts led to regulatory approvals: the US FDA approved oliceridine in August 2020 and China NMPA approved tegileridine (SHR8554) in January 2024 for treating moderate to severe acute pain [7].

However, the original hypothesis has been challenged. Kliewer and colleagues made mice with MOR that cannot recruit beta-arrestin due to mutation of phosphorylation sites. They found that while analgesia was improved and tolerance was reduced, respiratory depression and constipation were not better [8,9]. Gillis and colleagues offered a different explanation: the benefits of biased agonists may simply come from their lower intrinsic efficacy (partial agonism) rather than pathway selectivity [10]. If this is true, any MOR agonist with low enough efficacy would have fewer side effects at pain-relieving doses, regardless of signaling bias.

The debate continues. Stahl and Bohn reanalyzed the data and argued that bias factors remain important when calculated properly [11]. On the other hand, a recent

review by Breault and colleagues described this situation as the rise and fall of ligand biased signaling [12]. This uncertainty has important consequences: if the benefits of these newer drugs come from partial agonism rather than true bias, predictions about their physiological effects might be quite different.

## The autonomic gap

One area that has received little systematic attention is the effect of biased MOR agonists on the autonomic nervous system (ANS). Traditional opioids have well-known cardiovascular effects including slowed heart rate, low blood pressure and changes in heart rate variability (HRV). These effects occur mainly through central mechanisms, particularly through brainstem nuclei such as the nucleus tractus solitarius and nucleus ambiguus that control parasympathetic output to the heart [13,14]. While MOR has been found in cardiac tissue, its functional importance at the peripheral level is much less than at central sites.

These autonomic effects matter clinically. Opioid-induced hemodynamic instability creates problems during and after surgery. Changes in ANS function may also affect postoperative recovery through the cholinergic anti-inflammatory pathway, where vagal activity helps control systemic inflammation [15,16]. Whether biased MOR agonists produce different autonomic effects compared with traditional opioids has not been studied systematically.

A critical mechanistic uncertainty in this field is the "GIRK Paradox." Opioid-induced vagal enhancement is typically mediated via the disinhibition of GABAergic interneurons rather than direct excitation of vagal motor neurons. Because G-protein-gated inwardly rectifying potassium (GIRK) channels are entirely Gβγ subunit-dependent, robust G-protein signaling in brainstem nuclei (e.g., nucleus tractus solitarius) continues to modulate neuronal excitability. It remains an open question whether reduced β-arrestin recruitment alone is sufficient to spare these distinct autonomic pathways.

Furthermore, it is crucial to distinguish between receptor-engineered bias (e.g., using phosphorylation-deficient mutant mice) and ligand-driven bias (using novel drugs like oliceridine or tegileridine). While genetic models provide proof-of-concept, they may not perfectly translate to the pharmacological behavior of clinical drugs. Mapping how autonomic outcomes differ between these two distinct experimental approaches is a key focus of this review.

Several reasons suggest this question deserves investigation.

Mechanistically, evidence from other receptor systems indicates that beta-arrestin signaling plays a crucial role in cardiovascular regulation. For instance, carvedilol, a beta-blocker associated with superior heart failure outcomes, functions as a beta-arrestin-biased agonist [17], while the biased ligand TRV120067 has demonstrated cardioprotective effects in animal models [18]. These findings suggest that modulating arrestin recruitment could influence cardiovascular function through pathways extending beyond traditional opioid receptors.

In the clinical setting, Heart Rate Variability (HRV) has emerged as a robust prognostic biomarker, with reduced preoperative variability predicting adverse outcomes across various surgical specialties [15,19]. Consequently, if biased MOR agonists can preserve HRV integrity better than conventional opioids, this would represent a clinically significant advantage distinct from their established respiratory and gastrointestinal benefits.

Furthermore, existing regulatory data warrants re-examination through an autonomic lens. The FDA approval process for oliceridine generated substantial cardiovascular safety data, including two thorough QT studies that identified concentration-dependent QTc effects [20]. Given that the QT interval reflects a complex interplay between direct ion channel effects and autonomic tone, synthesizing this data is essential for a complete physiological profile.

Finally, emerging clinical trials underscore the relevance of this inquiry. A recent randomized study by Chen et al. comparing oliceridine with sufentanil in elderly hypertensive patients reported significantly improved hemodynamic stability with the biased agonist [21]. While this provides preliminary empirical support for autonomic advantages, a systematic synthesis of all available evidence remains necessary.

## Mechanistic paradigms

This scoping review aims to map existing evidence against three prevailing mechanistic paradigms regarding the autonomic effects of biased MOR agonists:

**Paradigm 1 (Bias Preservation):** G protein bias at MOR avoids beta-arrestin-mediated suppression of central parasympathetic output, resulting in preserved HRV and autonomic balance across all doses. If this is true, biased agonists should show good autonomic profiles even at high analgesic doses.

**Paradigm 2 (Efficacy Ceiling):** Any autonomic benefits come from low intrinsic efficacy rather than signaling bias. If this is true, HRV preservation should get worse at higher doses as receptor occupancy approaches maximum, and the autonomic profile should look similar to other partial agonists like buprenorphine.

**Paradigm 3 (Null):** No meaningful differences exist between biased agonists and traditional opioids for autonomic effects. If this is true, any observed hemodynamic differences come from pharmacokinetic factors like rapid onset and offset rather than signaling differences.

By mapping available evidence across drugs, doses, species and measurement times, this review can provide physiological data relevant to distinguishing among these paradigms, though it cannot give a definitive answer (Fig 1).

## Objectives

A scoping review methodology was deliberately chosen over a systematic review because our objective is to map a highly heterogeneous body of literature spanning the entire translational continuum (from in vitro molecular assays to in vivo animal models and human clinical trials). The diverse study designs, varying operational definitions of "biased agonism," and the exploratory nature of linking these to autonomic outcomes preclude the narrow PICO framework and quantitative meta-analysis required for a systematic review.

The main objective of this scoping review is to identify, characterize and put together existing evidence about the effects of G protein-biased MOR agonists on autonomic nervous system function. Secondary objectives include: mapping evidence gaps especially for tegileridine which has much less published data than oliceridine; mapping empirical data against the three mechanistic paradigms to clarify key concepts; and helping design future clinical trials that include autonomic endpoints.

## Methods

### Protocol design

This scoping review protocol follows the JBI methodology for scoping reviews as described by Peters and colleagues [22]. The completed review will be reported according to the PRISMA extension for Scoping Reviews (PRISMA-ScR) [23]. This protocol will be registered with the Open Science Framework before starting the search.

### Eligibility criteria

**Population/Exposure:** We will include studies examining G protein-biased or low-efficacy MOR agonists. The main drugs of interest are oliceridine (TRV130, Olinvyk) and tegileridine (SHR8554, Aisute), as these have regulatory approval. Other investigational biased MOR agonists will also be included: PZM21, TRV734, SR-17018 and novel compounds such as oxa-iboga alkaloids. Studies comparing these drugs to traditional opioids like morphine, fentanyl, hydromorphone, oxycodone and sufentanil are of particular interest. Studies using beta-arrestin-2 knockout or phosphorylation-deficient MOR models with cardiovascular outcomes will be included for mechanistic context. Studies examining only traditional opioids without comparison to biased agonists will be excluded.

**Concept:** The concept of interest is autonomic nervous system function. We draw a strict conceptual distinction between primary autonomic indices (which directly reflect neural tone) and secondary cardiovascular safety outcomes (which are composite measures). To keep things clear, outcomes will be divided into two categories. Primary autonomic indices include: heart rate variability (time-domain measures like RMSSD and SDNN; frequency-domain measures like LF, HF and LF/HF ratio; and nonlinear measures), baroreflex sensitivity, pupillometry as a parasympathetic indicator,

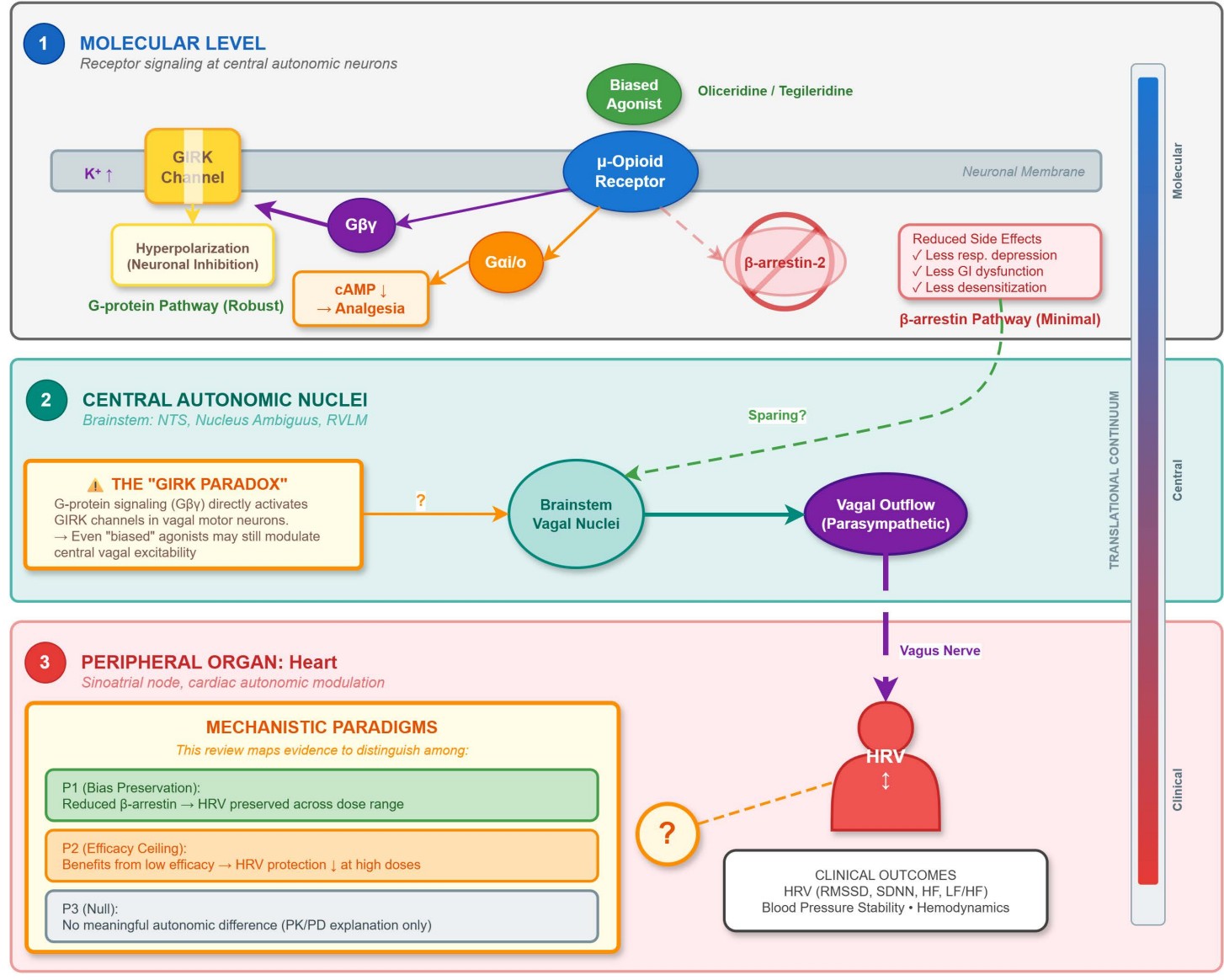

**Fig 1. Conceptual framework illustrating the translational pathway from mu-opioid receptor (MOR) signaling to autonomic outcomes.** The "GIRK Paradox" highlights that robust G-protein signaling may still modulate central vagal excitability, raising the question of whether reduced β-arrestin recruitment is sufficient to preserve autonomic homeostasis.

catecholamine levels as a sympathetic indicator, and direct measures of vagal or sympathetic nerve activity. Cardiovascular safety context includes: heart rate, blood pressure, QT/QTc interval, and cardiovascular adverse events such as bradycardia, hypotension and arrhythmia. These will be reported but interpreted with recognition that they reflect both direct cardiac effects and autonomic modulation.

**Context:** We will consider studies across the translational spectrum: in vitro and cell-based assays examining receptor signaling; in vivo animal studies of any species with stratification by anesthetic state (conscious versus anesthetized); healthy volunteer studies including pharmacokinetic and thorough QT studies; clinical trials in surgical or medical patients; and post-marketing surveillance studies. Regulatory documents from FDA and NMPA will be searched for unpublished

safety data. No restrictions on geographic setting or publication date will be applied. To allow meaningful comparison, measurement timepoints will be categorized as: baseline/pre-drug, acute (during drug or within 2 hours), early recovery (2–24 hours), and late recovery (more than 24 hours).

**Types of sources:** We will include original research: in vitro studies, animal experiments, phase I-IV clinical trials and observational studies. Reviews will be excluded from primary analysis but their reference lists will be checked for additional primary studies. Conference abstracts will be excluded due to lack of methodological detail. Case reports will be excluded unless they describe cardiovascular adverse events not captured in larger studies. There will be no language restrictions. Given that tegileridine was developed and approved in China, we will systematically search major Chinese databases (CNKI, Wanfang Data, and SinoMed) to ensure comprehensive capture of relevant clinical and preclinical data.

## Search strategy

The search strategy utilizes a three-block conceptual framework: (1) biased agonism and specific agents (e.g., "oliceridine," "tegileridine," "biased agonist," "low efficacy"); (2) mu-opioid receptor targets; and (3) autonomic/cardiovascular outcomes (e.g., "heart rate variability," "parasympathetic," "GIRK"). The full line-by-line search strategy for PubMed is provided in S2 Appendix.

We will search the following databases from inception through March 2026: PubMed including MEDLINE, Embase via Ovid, Web of Science Core Collection, and Cochrane Central Register of Controlled Trials. We will also search regulatory databases: FDA Drugs@FDA for oliceridine approval documents (NDA 210730), CDE/NMPA database for tegileridine documents, and ClinicalTrials.gov and ChiCTR for registered trials with cardiovascular outcomes. Reference lists of included studies will be checked. The FDA Adverse Event Reporting System (FAERS) will be queried for post-marketing cardiovascular safety signals.

## Study selection

Search results will be imported into Rayyan software and duplicates removed. Two reviewers (e.g., YZ, QC, or QZ) will independently screen titles and abstracts, followed by full-text review of potentially relevant articles. A calibration exercise using 30 randomly selected records will be done first to ensure consistent application of eligibility criteria (target agreement kappa 0.70 or higher). Disagreements will be resolved through discussion, with a third reviewer (HC or YkZ) available if needed. The screening process will be documented in a PRISMA flow diagram with reasons for full-text exclusions.

## Data extraction

A standardized data extraction form was developed for this review (S3 Appendix). The form was tested on three different types of studies (one preclinical, one clinical trial, one mechanistic) and improved accordingly. Data extraction will be done by one reviewer (YZ, QC, or YyZ) and checked by a second reviewer (HC or QZ).

**For all studies we will extract:** study identification information; study design and evidence level; drugs examined with doses and routes; comparators; sample size; funding source; and potential conflicts of interest.

**For preclinical studies:** species and strain, experimental model, anesthetic state with agent specified if used, autonomic and cardiovascular outcomes measured, measurement timepoints, and main findings with effect sizes where available.

**For clinical studies:** population characteristics, primary and secondary endpoints, cardiovascular adverse event rates, HRV or other autonomic data with measurement methods, dose-response relationships if reported, and measurement timepoints.

**For mechanistic studies:** receptor system, signaling pathway examined, and relevance to cardiovascular regulation. To address the heterogeneity of biased agonism, we will explicitly extract whether the "bias" in each study is ligand-driven (pharmacological) or receptor-engineered (genetic). Furthermore, we will extract bias factors (with their calculation

methods and reference ligands, e.g., DAMGO) and intrinsic efficacy values (e.g., Emax for G-protein activation). This will allow us to stratify drugs into "biased agonists" versus "low-efficacy/partial agonists" during data synthesis.

When relevant data are incomplete, we will contact authors (maximum two attempts). Cardiovascular safety data from regulatory submissions will be extracted from publicly available FDA briefing documents.

**Critical appraisal:** Although formal risk of bias assessment is optional in scoping reviews, we will conduct quality assessments due to the mechanistic controversies in this field. We will use the JBI Critical Appraisal Checklists for clinical studies, and the SYRCLE risk of bias tool for animal studies. For in vitro studies, we will assess the methodological rigor of bias factor calculations. This quality assessment will not be used to exclude studies but will inform the narrative synthesis to highlight whether conflicting results stem from methodological flaws or unmeasured confounders (e.g., inadequate anesthesia control).

## Data synthesis and presentation

Given the expected variety of evidence types, formal meta-analysis is not planned. Results will be synthesized in narrative form and organized using an Evidence Matrix with three dimensions:

**Dimension 1 (Intervention Type):** Oliceridine, tegileridine, other investigational compounds, and receptor-engineered models (e.g., β-arrestin knockout).

**Dimension 2 (Evidence level):** In vitro cellular assays, preclinical in vivo models, healthy volunteers, and patient populations.

**Dimension 3 (Timepoint):** Baseline, acute, early recovery, and late recovery.

To handle critical confounders, data will be synthesized using stratified analyses. Clinical data will be stratified by patient condition and concurrent anesthesia. Preclinical in vivo data will be strictly separated by anesthetic state (conscious with telemetry versus anesthetized), recognizing that anesthetic agents profoundly depress baseline autonomic tone.

The main visual output will be an Evidence Gap Map showing how studies are distributed across the matrix cells. This will immediately show where evidence is concentrated (expected: oliceridine clinical trials) and where gaps exist (expected: tegileridine autonomic data, conscious animal HRV studies).

A Conceptual Framework Figure will show the proposed pathways from MOR activation through central autonomic nuclei to cardiac effects, pointing out evidence gaps at each translational level.

To operationalize the mapping of evidence against the three mechanistic paradigms, we will use the following pre-defined criteria during synthesis:

**Paradigm 1 (Bias Preservation):** Studies demonstrating preserved HRV or autonomic function despite increasing doses of the biased agonist, specifically when correlated with in vitro data confirming high G-protein efficacy but low β-arrestin recruitment.

**Paradigm 2 (Efficacy Ceiling):** Studies showing that autonomic preservation diminishes at higher doses (where receptor occupancy is high), or studies demonstrating that the autonomic profile mirrors that of known low-efficacy partial agonists (e.g., buprenorphine).

**Paradigm 3 (Null):** Studies showing no significant difference in autonomic outcomes between biased agonists and conventional opioids when matched for analgesic efficacy or pharmacokinetic profiles.

We will utilize an effect direction plot alongside the Evidence Matrix to visually represent these findings. We explicitly acknowledge that assigning heterogeneous evidence to these paradigms is exploratory and hypothesis-generating, rather than definitive proof.

## Ethics and dissemination

Ethical approval is not needed for this scoping review of published literature and regulatory documents. Results will be published in a peer-reviewed journal, with Systematic Reviews as the target venue. We plan conference presentations at relevant anesthesiology and pharmacology meetings. The extracted data will be available as supplementary material. We expect that findings will help design clinical trials that include HRV or other autonomic endpoints.

## Discussion

We expect to find that clinical development programs for oliceridine collected cardiovascular safety data including QT measurements and hemodynamic monitoring, but that HRV was rarely measured as a specific endpoint. In fact, our preliminary search found that no published study has examined oliceridine and HRV together. For tegileridine, we expect much fewer published studies in English, with most data available only in regulatory submissions or Chinese-language literature. This asymmetry itself represents an important finding that will guide research priorities.

Based on our preliminary literature review, we expect the following pattern: clinical trials will show that biased agonists cause less hypotension and bradycardia than traditional opioids at equivalent analgesic doses, supporting some form of autonomic advantage. However, whether this comes from true pathway bias or low intrinsic efficacy may remain unclear without dose-response data at high receptor occupancy. Preclinical studies, especially those in conscious animals with telemetry monitoring, may give more mechanistic insight but are probably limited in number.

The importance of this review goes beyond putting literature together. Enhanced Recovery After Surgery (ERAS) protocols increasingly emphasize hemodynamic stability and early mobilization. If some opioids preserve autonomic function better than others, this could affect analgesic choice in perioperative care. Also, if HRV turns out to be a useful distinguishing endpoint, future trials could include it to better characterize the physiological profiles of newer analgesics.

The inflammatory reflex pathway, where vagal tone affects systemic inflammation, provides additional reason: ANS-preserving analgesics might offer benefits for postoperative inflammation and recovery that go beyond hemodynamic considerations.

Finally, this review will specifically map the evidence gap for tegileridine compared with oliceridine. As tegileridine enters clinical use in China and potentially other markets, setting research priorities for characterizing its autonomic profile becomes timely. Our planned randomized controlled trial examining autonomic outcomes with tegileridine will directly address gaps identified by this review.

## Limitations

Several limitations should be noted. First, although we include major Chinese databases to capture tegileridine data, relevant studies published in other non-English languages may still be missed. Second, variety across study designs, populations and outcome measures will prevent quantitative synthesis. Third, the mechanistic paradigms cannot be definitively validated by observational evidence; experimental studies with dose-response designs would be needed. Fourth, publication bias may favor positive findings. Fifth, the ongoing debate about how to calculate bias factors creates uncertainty when comparing across studies.

## Supporting information

**S1 Checklist. PRISMA-P 2015 checklist.**
(DOCX)

**S2 Checklist. PRISMA-ScR checklist.**
(DOCX)

**S1 Appendix. Search strategy.**
(DOCX)

**S2 Appendix. Data extraction form.**
(DOCX)

**S3 Appendix. PCC framework summary.**
(DOCX)

## Author contributions

**Conceptualization:** Yi Zhang, Qiuxiang Chen, Yuanyuan Zhou.

**Data curation:** Qingjun Zeng, Haishan Cui.

**Formal analysis:** Haishan Cui.

**Funding acquisition:** Yi Zhang.

**Investigation:** Qingjun Zeng.

**Methodology:** Yi Zhang, Qiuxiang Chen, Yuanyuan Zhou.

**Supervision:** Yukai Zhou.

**Writing – original draft:** Yi Zhang, Qiuxiang Chen, Yuanyuan Zhou.

**Writing – review & editing:** Yi Zhang, Qiuxiang Chen, Yuanyuan Zhou, Yukai Zhou.

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
