## [Decision Letter · Decision Letter 0]

13 Apr 2026

PONE-D-26-11896Autonomic nervous system modulation by G protein-biased mu-opioid receptor agonists: a translational scoping review protocolPLOS One

Dear Dr. Yukai,

Thank you for submitting your manuscript to PLOS ONE. After careful consideration, we feel that it has merit but does not fully meet PLOS ONE’s publication criteria as it currently stands. Therefore, we invite you to submit a revised version of the manuscript that addresses the points raised during the review process.

We look forward to receiving your revised manuscript.

Kind regards,

Yusuf Oloruntoyin Ayipo, Ph.D

Academic Editor

PLOS One

Journal Requirements:

5.If the reviewer comments include a recommendation to cite specific previously published works, please review and evaluate these publications to determine whether they are relevant and should be cited. There is no requirement to cite these works unless the editor has indicated otherwise

Additional Editor Comments:

The submission reflects scientific relevance. However, some fundamental concerns have been raised by the reviewers affecting pivotal sections of the submission. Kindly pay close attention to the queries of the reviewers and address them critically before resubmission.

Reviewers' comments:

Reviewer's Responses to Questions

**Comments to the Author**

1. Does the manuscript provide a valid rationale for the proposed study, with clearly identified and justified research questions?

Reviewer #1: Yes

Reviewer #2: Partly

Reviewer #3: Yes

Reviewer #4: Yes

Reviewer #5: Yes

2. Is the protocol technically sound and planned in a manner that will lead to a meaningful outcome and allow testing the stated hypotheses?

Reviewer #1: Yes

Reviewer #2: Partly

Reviewer #3: Yes

Reviewer #4: Partly

Reviewer #5: Yes

3. Is the methodology feasible and described in sufficient detail to allow the work to be replicable?

Reviewer #1: Yes

Reviewer #2: Yes

Reviewer #3: Yes

Reviewer #4: Yes

Reviewer #5: Yes

4. Have the authors described where all data underlying the findings will be made available when the study is complete?

Reviewer #1: Yes

Reviewer #2: Yes

Reviewer #3: Yes

Reviewer #4: Yes

Reviewer #5: Yes

5. Is the manuscript presented in an intelligible fashion and written in standard English?

Reviewer #1: Yes

Reviewer #2: No

Reviewer #3: Yes

Reviewer #4: Yes

Reviewer #5: Yes

6. Review Comments to the Author

You may also provide optional suggestions and comments to authors that they might find helpful in planning their study.

Reviewer #1: Review’s Comments

1. Although the study follows JBI and PRISMA-ScR guidelines, the protocol remains largely descriptive and lacks deeper methodological rigor (e.g., no clear plan for handling heterogeneity, bias assessment, or data quality weighting).

2. The protocol does not specify whether included studies will be assessed for quality or risk of bias, which weakens the interpretability and credibility of the final synthesis.

3. The study includes preclinical, clinical, mechanistic, and regulatory data all together, which introduces substantial heterogeneity and may limit meaningful interpretation or conclusions.

4. While the authors state that meta-analysis is not planned, there is no alternative structured approach (e.g., semi-quantitative synthesis, vote counting, or effect direction analysis), making the output potentially weak.

5. The three paradigms (bias preservation, efficacy ceiling, null) are interesting but not operationalized there are no predefined criteria for assigning evidence to each paradigm.

6. Exclusion of non-English studies (despite tegileridine being developed in China) introduces bias and may omit critical data.

7. The authors justify a scoping review due to heterogeneity, but the research question is sufficiently focused that a systematic review (or hybrid approach) could be more appropriate.

8. Scoping reviews on opioid pharmacology and safety are common; the novelty here (ANS focus) is interesting but not strongly positioned as a significant gap.

9. The search strategy is mentioned but not fully detailed in the main text, which affects reproducibility and transparency.

10. The distinction between autonomic indices and cardiovascular outcomes is acknowledged but not clearly resolved, which may lead to inconsistent interpretation.

11. Given the known controversy in biased agonism, the protocol does not clearly state how contradictory findings will be synthesized or interpreted.

12. The proposed “Evidence Matrix” is conceptually useful but lacks detail on how it will be constructed, analyzed, or interpreted.

13. Important confounders (dose, pharmacokinetics, patient condition, anesthesia effects) are acknowledged but not formally incorporated into analysis strategy.

Reviewer #2: 1- authors should clarify why they opt to select scoping review rather than having it as systematic review, in particular when they reffered to "wide range of relevant literature," but they also reported that their preliminary search found no published study examining oliceridine and HRV together.

2- Authors should make clear justification how the GIRK paradox and bias factor heterogeneity will be handled in data extraction and synthesis. its also recommended that figure 1 caption could be shortened as some of the mechanistic detail it contains would fit better within the text.

3- Authors should explicitly address how the English-language restriction may affect the depth and completeness of evidence available for tegileridine compared with oliceridine.

Reviewer #3: I still question the decision to only include English language articles given the presence of these drugs in China. However this was identified as a bias.

Reviewer #4: 1. Interpretation of Mechanistic Evidence on Biased Signaling

The conceptual framing of G protein-biased μ-opioid receptor agonism would benefit from a clearer distinction between ligand-driven (pharmacological) bias and receptor-engineered (genetic) bias. For example, the widely cited study by Kliewer et al. (2019) does not evaluate biased agonist drugs. Instead, it employs phosphorylation-deficient μ-opioid receptors to experimentally enforce G protein-biased signaling using conventional opioids such as morphine and fentanyl. This distinction is critical because findings from receptor-modified genetic models—where β-arrestin recruitment is disrupted—may not translate directly to the pharmacological behavior of clinically relevant biased ligands (e.g., oliceridine/TRV130). Notably, adverse effects were not mitigated (and in some cases worsened) in these models, which challenges the assumption that G protein bias alone is sufficient to improve the autonomic nervous system or side-effect profile of μ-opioid receptor agonists.

The protocol would therefore benefit from explicitly stating how mechanistic studies using receptor-modified systems will be interpreted relative to ligand-based biased agonism, and whether such studies will be analyzed separately (or stratified) in the evidence synthesis.

2. A second related concern is that the inclusion criterion (under Population/Exposure) grouping “G protein-biased or low-efficacy MOR agonists” conflates two mechanistically distinct pharmacological concepts. While these categories may produce overlapping physiological outcomes, they differ fundamentally: one concerns signaling selectivity (bias), while the other concerns intrinsic efficacy. Combining them without clear operational definitions or planned stratification risks substantial heterogeneity and may limit the interpretability of findings. The authors should clarify how these classes will be defined, distinguished, and analyzed (e.g., via subgroup or sensitivity analyses) within the review.

3. Given that G protein-biased agonism is fundamentally defined and quantified at the receptor signaling level (typically via in vitro assays), the current protocol’s apparent limited emphasis on in vitro or mechanistic studies may constrain the ability to properly contextualize translational findings on autonomic nervous system modulation.

4. Finally, clarification is needed on how heterogeneous evidence sources (e.g., controlled trials, pharmacokinetic studies, and post-marketing surveillance) will be categorized and synthesized, particularly when study designs and outcome measures vary widely.

Reviewer #5: This manuscript presents a well-structured and methodologically sound protocol for a scoping review investigating the autonomic nervous system effects of G protein-biased μ-opioid receptor agonists. The topic is timely, relevant, and addresses an important gap in the literature, particularly given the growing clinical use of agents such as oliceridine and tegileridine.

The protocol follows established methodological frameworks (JBI and PRISMA-ScR), includes a clear search strategy, predefined eligibility criteria, and an organized data extraction and synthesis plan. The proposed evidence matrix and translational approach are strengths.

However, there are several important concerns that should be addressed:

Conceptual clarity: The manuscript attempts to relate heterogeneous evidence (mechanistic, preclinical, and clinical) to mechanistic paradigms (bias vs intrinsic efficacy). This may overreach the capabilities of a scoping review. The authors should clearly state that such interpretations are hypothesis-generating rather than definitive.

Definition of biased agonism: The inclusion criteria combine biased agonists with low-efficacy/partial agonists without a clear operational definition. This requires clarification or stratification.

Synthesis approach: The protocol should better distinguish between different levels of evidence (molecular, animal, clinical) to avoid overinterpretation.

Language restriction bias: Limiting to English-language studies may exclude relevant data, especially for tegileridine. This should be justified more clearly.

Quality assessment: The absence of any plan for study quality or risk-of-bias assessment should be justified.

Overall, the study is promising but requires clarification and refinement to ensure methodological and conceptual rigor.

7. PLOS authors have the option to publish the peer review history of their article (what does this mean?). If published, this will include your full peer review and any attached files.

Reviewer #1: **Yes:** Favour Ezeogu

Reviewer #2: **Yes:** Waleed A Alananzeh

Reviewer #3: No

Reviewer #4: No

Reviewer #5: No

---

## [Author Response · Author response to Decision Letter 1]

13 Apr 2026

Response to Reviewers

Dear Dr. Yusuf Oloruntoyin Ayipo,

Thank you for giving us the opportunity to submit a revised draft of our manuscript. We deeply appreciate the time and effort you and the five reviewers have dedicated to evaluating our protocol. The reviewers’ comments, particularly regarding the inclusion of in vitro studies, the distinction between ligand-driven and receptor-engineered bias, and the removal of language restrictions, were incredibly insightful. Incorporating these suggestions has significantly improved the methodological rigor and clarity of our protocol.

We have carefully addressed every comment raised by the reviewers. Please find our point-by-point responses below. The corresponding changes have been highlighted in the "Revised Manuscript with Track Changes" file.

Response to Journal Requirements

1.PRISMA-P checklist requirement.

Response: Thank you for this guidance. We have completed the PRISMA-P 2015 checklist and uploaded it as a separate Supporting Information file (S1 Checklist). Since this is a scoping review protocol, we have marked items exclusively pertaining to systematic reviews/meta-analyses (e.g., risk of bias assessment, meta-biases) as "N/A", while providing the relevant page numbers for all other items.

2.Removal of funding information from the manuscript.

Response: We have removed the "Funding" section from the main text of the manuscript. The funding information is now only provided in the online submission form.

3.ORCID iD validation.

Response: The corresponding author (Yukai Zhou) has successfully linked and validated his ORCID iD in the Editorial Manager system.

4.Captions for Supporting Information files.

Response: We have added a "Supporting information" section at the end of the manuscript (after the References) with clear captions for all supplementary files, and we have updated all in-text citations accordingly (e.g., changing "Appendix 1" to "S2 Appendix").

Response to Reviewer #1

Comment 1.1 & 1.2: Lack of deeper methodological rigor and no clear plan for handling heterogeneity, bias assessment, or data quality weighting.

Response: We completely agree with your concern. Initially, we omitted a formal quality assessment because it is optional in scoping reviews. However, given the mechanistic controversies in this field, we realize that assessing study quality is crucial. We have now added a "Critical appraisal" section. We will use the JBI Critical Appraisal Checklists for clinical studies and the SYRCLE risk of bias tool for animal studies. For in vitro studies, we will assess the methodological rigor of bias factor calculations.

Action taken: A new "Critical appraisal" section has been added to the Methods (Page 13).

Comment 1.3: Including preclinical, clinical, mechanistic, and regulatory data together introduces substantial heterogeneity.

Response: Thank you for pointing this out. We acknowledge the high heterogeneity. To prevent meaningless pooling of data, we will strictly stratify our synthesis. We have updated the "Data synthesis and presentation" section to clarify that data will be synthesized within their specific evidence levels (e.g., in vitro, animal, human) rather than across them.

Action taken: Clarified in the "Data synthesis and presentation" section (Page 14).

Comment 1.4: No alternative structured approach (e.g., semi-quantitative synthesis, vote counting, or effect direction analysis) since meta-analysis is not planned.

Response: This is a very helpful suggestion. We have added a plan to use an "effect direction plot" alongside our Evidence Matrix. This will visually represent the direction of autonomic effects (e.g., HRV preserved, HRV reduced, or conflicting) across different drug doses and study types.

Action taken: Added to the "Data synthesis and presentation" section (Page 15).

Comment 1.5: The three paradigms (bias preservation, efficacy ceiling, null) are not operationalized; there are no predefined criteria.

Response: We apologize for the previous vagueness. We have now provided explicit, predefined criteria for how we will assign the extracted evidence to each of the three mechanistic paradigms.

Action taken: Detailed operational definitions for Paradigm 1, 2, and 3 have been added to the "Data synthesis and presentation" section (Pages 14-15).

Comment 1.6: Exclusion of non-English studies introduces bias and may omit critical data (especially for tegileridine).

Response: We completely agree. As clinical researchers from China, we initially limited the search to English to match standard international review practices, but we realize this would miss crucial data for tegileridine. We have removed the language restriction and officially added major Chinese databases (CNKI, Wanfang Data, and SinoMed) to our search strategy.

Action taken: Updated in the "Types of sources" (Page 11), "Limitations" (Page 16), and the search strategy appendices.

Comment 1.7: The research question is sufficiently focused that a systematic review could be more appropriate. Why a scoping review?

Response: Thank you for this methodological question. We chose a scoping review because we are mapping a highly heterogeneous body of literature that spans the entire translational continuum (from in vitro cell assays to in vivo animal models and human trials). The varying operational definitions of "biased agonism" and the exploratory nature of linking these to autonomic outcomes preclude the narrow PICO framework required for a systematic review. We have added this justification to the manuscript.

Action taken: Added a rationale paragraph under "Objectives" (Page 9).

Comment 1.8: ANS focus is interesting but not strongly positioned as a significant gap.

Response: We have strengthened the Introduction to better highlight the clinical and mechanistic gap. Specifically, we introduced the "GIRK Paradox" (G-protein-gated inwardly rectifying potassium channels), explaining that since vagal modulation is largely G-protein dependent, it remains highly uncertain whether reducing β-arrestin alone can preserve autonomic homeostasis.

Action taken: Added mechanistic details to the "The autonomic gap" section in the Introduction (Page 6).

Comment 1.9: The search strategy is mentioned but not fully detailed in the main text.

Response: We have expanded the description of the search strategy in the main text to clearly show the three-block conceptual framework we used.

Action taken: Updated the "Search strategy" section (Page 11).

Comment 1.10: Distinction between autonomic indices and cardiovascular outcomes is acknowledged but not clearly resolved.

Response: We have refined our definitions to draw a strict conceptual distinction. We now explicitly state that primary autonomic indices (e.g., HRV, pupillometry) directly reflect neural tone, while secondary cardiovascular outcomes (e.g., blood pressure, QT interval) are composite measures influenced by both autonomic tone and direct cardiac effects.

Action taken: Clarified in the "Concept" section of the Eligibility criteria (Page 10).

Comment 1.11: Protocol does not clearly state how contradictory findings will be synthesized or interpreted.

Response: We have clarified that our newly added quality assessment (critical appraisal) and the effect direction plot will be used specifically to explore whether contradictory findings stem from methodological flaws or unmeasured confounders (such as different anesthetic agents used in animal models).

Action taken: Addressed in the "Critical appraisal" and "Data synthesis" sections (Pages 13-14).

Comment 1.12: The proposed “Evidence Matrix” lacks detail on how it will be constructed.

Response: We have provided specific details on the axes of the Evidence Matrix: Dimension 1 (Intervention Type), Dimension 2 (Evidence Level), and Dimension 3 (Timepoint).

Action taken: Detailed in the "Data synthesis and presentation" section (Pages 13-14).

Comment 1.13: Important confounders (dose, PK, patient condition, anesthesia effects) are not formally incorporated into analysis strategy.

Response: We agree that these confounders are critical. We have added a specific plan to synthesize data using stratified analyses. For example, clinical data will be stratified by patient condition and concurrent anesthesia, and preclinical data will be strictly separated by anesthetic state (conscious vs. anesthetized).

Action taken: Added to the "Data synthesis and presentation" section (Page 14).

Response to Reviewer #2

Comment 2.1: Clarify why opt to select scoping review rather than systematic review.

Response: Thank you for the comment. As also requested by Reviewer 1, we have added a clear justification. The inclusion of in vitro assays, animal models, and clinical trials creates a translational spectrum that is too broad for a traditional systematic review. A scoping review allows us to map these diverse evidence types to identify gaps (especially for tegileridine) before narrower systematic reviews can be conducted.

Action taken: Added to the "Objectives" section (Page 9).

Comment 2.2: Make clear justification how the GIRK paradox and bias factor heterogeneity will be handled. Figure 1 caption could be shortened.

Response: We greatly appreciate this advice. We have significantly shortened the caption for Figure 1 and moved the detailed explanation of the "GIRK Paradox" into the Introduction text. Furthermore, to handle bias factor heterogeneity, we will explicitly extract the calculation methods and reference ligands (e.g., DAMGO vs. morphine) used in each study.

Action taken: Figure 1 caption shortened (Page 8). GIRK paradox moved to Introduction (Page 6). Bias factor extraction detailed in "Data extraction" (Page 13).

Comment 2.3: Address how the English-language restriction may affect the depth and completeness of evidence for tegileridine.

Response: We agree this was a major limitation. We have completely removed the English-language restriction and added major Chinese databases (CNKI, Wanfang Data, SinoMed) to our search strategy to ensure we capture all available data for tegileridine.

Action taken: Updated in "Types of sources" (Page 11) and "Limitations" (Page 16).

Response to Reviewer #3

Comment 3.1: I still question the decision to only include English language articles given the presence of these drugs in China.

Response: We sincerely thank the reviewer for pointing out this critical flaw. We have entirely removed the language restriction. We will systematically search Chinese databases (CNKI, Wanfang Data, and SinoMed) to ensure no tegileridine data is missed.

Action taken: Search strategy and limitations updated accordingly (Pages 11, 16).

Response to Reviewer #4

Comment 4.1: Clearer distinction needed between ligand-driven (pharmacological) bias and receptor-engineered (genetic) bias (e.g., Kliewer et al. 2019). State how they will be interpreted and analyzed separately.

Response: We are very grateful for this expert insight. We completely agree that findings from phosphorylation-deficient mutant mice cannot be directly translated to the clinical effects of biased drugs. We have added a paragraph in the Introduction to explicitly distinguish between ligand-driven and receptor-engineered bias. Furthermore, we have updated our data extraction form to capture the "bias source", and we will stratify these two distinct experimental approaches in our Evidence Matrix.

Action taken: Added distinction in Introduction (Page 6). Updated "Data extraction" (Page 13) and "Data synthesis" (Page 14).

Comment 4.2: Grouping “G protein-biased or low-efficacy MOR agonists” conflates two mechanistically distinct pharmacological concepts. Clarify how these classes will be defined and distinguished.

Response: You are absolutely correct. Bias and intrinsic efficacy are fundamentally different concepts, and combining them was a conceptual error in our previous draft. To resolve this, we will explicitly extract intrinsic efficacy values (e.g., Emax for G-protein activation) and bias factors. During data synthesis, we will use these extracted values to stratify the drugs into "biased agonists" versus "low-efficacy/partial agonists" to see if their autonomic profiles differ.

Action taken: Clarification added to the "Data extraction" section (Page 13).

Comment 4.3: Limited emphasis on in vitro or mechanistic studies may constrain the ability to properly contextualize translational findings.

Response: We appreciate this important feedback. As clinical researchers, we initially excluded in vitro studies to keep the review manageable. However, we agree that without in vitro signaling data, we cannot accurately define "bias" or "efficacy." We have now officially included in vitro cellular assays in our eligibility criteria.

Action taken: In vitro studies added to "Eligibility criteria" (Pages 9, 10, 11).

Comment 4.4: Clarification is needed on how heterogeneous evidence sources (e.g., controlled trials, PK studies, post-marketing) will be categorized and synthesized.

Response: We have refined our "Data synthesis and presentation" section. We will use an Evidence Matrix to strictly separate the data by Evidence Level (Dimension 2: in vitro, preclinical in vivo, healthy volunteers, patient populations). We will not pool data across these different study designs.

Action taken: Detailed in the "Data synthesis and presentation" section (Page 14).

Response to Reviewer #5

Comment 5.1: Conceptual clarity: Relating heterogeneous evidence to mechanistic paradigms may overreach a scoping review. State that such interpretations are hypothesis-generating rather than definitive.

Response: We agree with your assessment. A scoping review cannot definitively prove a pharmacological mechanism. To avoid overreaching, we have changed the term "competing hypotheses" to "mechanistic paradigms". We have also added an explicit statement acknowledging that assigning heterogeneous evidence to these paradigms is purely exploratory and hypothesis-generating.

Action taken: Terminology updated throughout the manuscript. Explicit disclaimer added at the end of the "Data synthesis and presentation" section (Page 15).

Comment 5.2: Definition of biased agonism: The inclusion criteria combine biased agonists with low-efficacy/partial agonists without a clear operational definition.

Response: As also suggested by Reviewer 4, we have corrected this conflation. We will extract specific bias factors and intrinsic efficacy measures (Emax) to operationally distinguish true biased agonists from low-efficacy partial agonists during the data synthesis phase.

Action taken: Added to the "Data extraction" section (Page 13).

Comment 5.3: Synthesis approach: The protocol should better distinguish between different levels of evidence (molecular, animal, clinical) to avoid overinterpretation.

Response: We have updated the Evidence Matrix description to ensure that molecular, preclinical, and clinical data are strictly stratified and synthesized within their own evidence levels.

Action taken: Clarified in the "Data synthesis and presentation" section (Page 14).

Comment 5.4: Language restriction bias: Limiting to English-language studies may exclude relevant data, especially for tegileridine.

Response: We have removed the English-language restriction and included Chinese databases (CNKI, Wanfang, SinoMed) in our search strategy.

Action taken: Updated in "Types of sources" (Page 11).

Comment 5.5: Quality assessment: The absence of any plan for study quality or risk-of-bias assessment should be justified.

Response: We agree that given the controversies in this field, quality assessment is necessary. We have added a "Critical appraisal" section, utilizing JBI tools for clinical studies and SYRCLE for animal studies, to inform our narrative synthesis.

Action taken: Added "Critical appraisal" section (Page 13).

Once again, we thank the Academic Editor and the Reviewers for their highly constructive comments, which have undoubtedly elevated the quality of our protocol. We hope the revised manuscript is now suitable for publication in PLOS ONE.

Sincerely,

Yukai Zhou, MD

Corresponding Author

---

## [Decision Letter · Decision Letter 1]

4 May 2026

Autonomic nervous system modulation by G protein-biased mu-opioid receptor agonists: a translational scoping review protocol

PONE-D-26-11896R1

Dear Dr. Yukai,

We’re pleased to inform you that your manuscript has been judged scientifically suitable for publication and will be formally accepted for publication once it meets all outstanding technical requirements.

Kind regards,

Yusuf Oloruntoyin Ayipo, Ph.D

Academic Editor

PLOS One

Additional Editor Comments (optional):

The submission is scientifically sound for publication in this title, and all the concerns raised by the respective reviewers regarding the manuscript quality have been satisfactorily addressed. I hereby recommend the manuscript for publication in the current version.

Reviewers' comments:

Reviewer's Responses to Questions

**Comments to the Author**

1. Does the manuscript provide a valid rationale for the proposed study, with clearly identified and justified research questions?

Reviewer #2: Yes

Reviewer #5: Yes

2. Is the protocol technically sound and planned in a manner that will lead to a meaningful outcome and allow testing the stated hypotheses?

Reviewer #2: Yes

Reviewer #5: Yes

3. Is the methodology feasible and described in sufficient detail to allow the work to be replicable?

Reviewer #2: Yes

Reviewer #5: Yes

4. Have the authors described where all data underlying the findings will be made available when the study is complete?

Reviewer #2: Yes

Reviewer #5: Yes

5. Is the manuscript presented in an intelligible fashion and written in standard English?

Reviewer #2: Yes

Reviewer #5: Yes

6. Review Comments to the Author

You may also provide optional suggestions and comments to authors that they might find helpful in planning their study.

Reviewer #2: I would like to thank the authors for their diligent efforts in addressing the requested revisions. Based on the thoroughness of their responses and the improvements made, I believe this manuscript is now ready for publication in PLOS

Reviewer #5: Thank you for the opportunity to review the revised version of this manuscript.

The authors have adequately addressed the comments and concerns raised in the initial review. The revisions made have improved the clarity, rigor, and overall quality of the manuscript, and the responses provided are satisfactory.

Based on the revisions, I consider the manuscript suitable for publication.

Sincerely,

7. PLOS authors have the option to publish the peer review history of their article (what does this mean?). If published, this will include your full peer review and any attached files.

Reviewer #2: **Yes:** Waleed A Alananzeh

Reviewer #5: No

---

## [Editor Report · Acceptance letter]

PONE-D-26-11896R1

PLOS One

Dear Dr. Zhou,

I'm pleased to inform you that your manuscript has been deemed suitable for publication in PLOS One. Congratulations! Your manuscript is now being handed over to our production team.

Kind regards,

on behalf of

Dr. Yusuf Oloruntoyin Ayipo

Academic Editor

PLOS One